# Post-Traumatic Trigeminal Neuropathic Pain after Dental Implant Surgery and the Injustice Experience Questionnaire

**Souichirou Tadokoro [1], Keita Takizawa [1], Kana Ozasa [1], Akiko Okada-ogawa [1], Yasuhide Kaneko [2], Jumi Nakata [3] and Noboru Noma [1,*]**

1   Department of Oral Medicine, Nihon University School of Dentistry, Tokyo 101-8310, Japan
2   Kaneko Dental Office, Omiya 321-0963, Japan
3   Division of Internal Medicine, Towa Hospital, Tokyo 120-0003, Japan
*   Correspondence: noma.noboru@nihon-u.ac.jp

**Abstract:** Painful post-traumatic trigeminal neuropathy (PTTN) is a known complication of dental implant therapy. Patients with PTTN develop sensory abnormalities in the orofacial region, which may be a psychosocial aspect, and dentists should assess somatosensory testing and psychosocial factors. The patients were assessed using quantitative sensory testing (QST). A 64-year-old female presented with allodynia of the left lower lip that occurred after a surgical implant procedure. Persistent pain started 4 months after the placement of two dental implants in the mandible. Sensory testing of these areas revealed warm hyposensitivity and mechanical hypersensitivity of the mandibular region. We also assessed PTTN-related perceived injustice using the Injustice Experience Questionnaire. The patient refused medication therapy such as pregabalin; therefore, autogenic training was adopted as an alternative management strategy. We conclude that for expensive dental procedures, such as implant placement, sufficient consensus should be obtained preoperatively before proceeding with surgery.

**Keywords:** PTTN; dental implant surgery; QST; IEQ; AT

## 1. Introduction

Implants are used to reconstruct compromised occlusions, leading to an improved quality of life and diet [1]. However, painful post-traumatic trigeminal neuropathy (PTTN) after dental implant surgery rapidly reduces the quality of life. Regarding the developmental pathology of PTTN, the interactions between the peripheral sensory nerves and the central nervous system (CNS) are highly complex. A severe injury can cause paresthesia and neuropathic pain, causing persistent discomfort, especially at night, and severely affecting the quality of life [2]. If the patient is not informed of the potential sensory deficits before implant surgery, not only will they feel an injustice has been done, but the CNS will also be affected, which may lead to the worsening of the sensory deficits.

Oral analgesics are one of the main treatments for many types of pain, because there are more specific pharmacologic protocols for neuropathic pain [3]. However, using pharmaceuticals for pain relief can result in unwanted side effects. Nonpharmacological psychological interventions have been used for pain management, such as progressive muscle relaxation, biofeedback, and autogenic training (AT).

AT is a self-relaxation procedure developed by Schultz approximately 100 years ago that applies passive concentration to specific combinations of psychophysiologically adapted stimuli [4]. This is addressed using six suggestive formulas aimed at increasing the relaxation and balance between sympathetic and parasympathetic control [5]. Relaxation has been suggested to affect pain by reducing tissue oxygen demand, breaking down lactic acid, reducing skeletal muscle tension and anxiety, and releasing endorphins [6].

Sullivan et al. [7] developed the Injustice Experience Questionnaire (IEQ) to measure perceived injustice related to musculoskeletal injuries. However, perceived injustice not

only occurs in the context of injury but is also present in chronic pain states with more insidious onsets [7]. Perceived injustice among patients with work-related disorders, particularly musculoskeletal conditions, correlates with pain severity, devastating conditions, fear of exercise, perception of disability, and depression [8]. High IEQ scores predict the failure to return to work and have been suggested to correlate with current unemployment rates [8]. Since implant treatment is expensive in the dental field, treatment failure may cause a sense of unfairness and chronic pain; however, there are no reports yet.

Here, we assessed pain using the Japanese version of the IEQ and measured somatosensory function after dental implant surgery, using a standardized quantitative sensory test (QSTs) developed by the German Research Network on Neuropathic Pain (DFNS).

## 2. Case

The patient was a 64-year-old female presenting with numbness in the left lower lip and chin.

Dental implant surgery for the left lower second premolar and second molar had been performed by a dental practitioner one year and three months ago. The patient had undergone laser treatment at the same dental clinic. Four months after the dental implant placement, the patient complained of abnormal sensations and discomfort when eating hot soba noodles and presented with a painful sensation in our clinic. The pain intensity was usually mild (3–4 on a visual analog scale, in which 0 = no pain and 10 = the worst pain). However, the patient described occasional exacerbations of moderate pain (5 or 6 on the same visual analog scale). She described the pain as a steady ache of mild-to-moderate intensity with occasional exacerbations of throbbing and pressure pain.

In our pain clinic, the patient strongly complained that she had not heard from her previous doctor about the possibility of such abnormal sensations in the chin or lower lip after implant surgery. A comprehensive examination of the intraoral and cranial nerves and a physical examination of the masticatory and cervical muscles were performed. Simple neurosensory testing (semiquantitative chairside techniques) revealed hyperesthesia in the chin and lower lip. A panoramic radiograph showed compression of the mandibular canal by a dental implant of the mandibular left second premolar.

### 2.1. Quantitative Sensory Testing (QST)

The QST protocol measures a wide range of sensory parameters [6], including the cold detection threshold (CDT), warm detection threshold (WDT), thermal sensory limen (TSL: the difference between the thresholds for alternating cool and warm detection), cold pain threshold (CPT), and heat pain threshold (HPT), which were measured using a thermal sensory testing device (TSA 2001-II, MEDOC, Tel Aviv, Israel) made with a Peltier element thermode and a $10 \times 10$ mm contact surface. The mechanical detection threshold (MDT), mechanical pain threshold and sensitivity (MPT and MPS), wind-up ratio (WUR: pain after repetitive pinprick stimulation), dynamic mechanical allodynia (DMA), pressure pain threshold (PPT), and vibration detection threshold (VDT) were measured using a standardized set.

### 2.2. Quantitative Sensory Testing Result

The QST protocol of the German Research Network on Neuropathic Pain was followed [9]. The absolute data on the intact side showed that CDT = −1.67, WDT = 2.07, TSL = 0.68, CPT = 14.4, HPT = 46.4, MDT = 0.245, MPT = 128, VDT = 6.83, PPT = 0.78, and WUR = 1.13. In contrast, the absolute data showed that CDT = −1.96, WDT = 14.3, TSL = 2.6, CPT = 15.6, HPT = 46.9, MDT = 0.66, MPT = 21.1, VDT = 6.5, PPT = 0.79, and WUR = 1.39. The abnormal values for the patient were for the warm detection threshold (WDT = 15.6 °C) and mechanical detection threshold (MDT = 0.65 mN). The MDT and WDT values indicated a gain of function on the affected side [9].

### 2.3. Injustice Experience Questionnaire

The patient's pain was evaluated using the IEQ, which was translated and back-translated into English and Japanese [10]. The IEQ's translation into Japanese was based on guidelines for cross-cultural adaptation of self-report measures. We employed the Japanese version of the IEQ. The IEQ is a self-reported measure of injury-related perceived injustice. The IEQ consists of 12 statements assessing the responder's appraisals of their injury and its consequences in terms of unfairness (e.g., "It all seems so unfair."), irreparability of loss ("My life will never be the same."), and attribution of blame ("I am suffering because of someone else's negligence."). The patient assessed the frequency with which they had experienced each of the 12 pain-related perceptions. The IEQ employs a 5-point scale from 0 to 4 as follows: never (0), rarely (1), sometimes (2), often (3), and all the time (4). The total score was calculated by adding all the items (theoretical range, 0–48). In addition, the proportion of patients who endorsed each item was calculated. Endorsement was defined as a rating of 'sometimes' or greater ($\geq 2$). The patient's total IEQ score was 26/48.

### 2.4. Management

Previously, the patients had experienced side effects to pharmacological management, such as pregabalin or amitriptyline, and our behavioral medicine team introduced AT. The AT was offered to the patient as an alternative treatment option and informed consent was obtained. Then, the effect of the VAS score after the AT was investigated.

Pain intensity was reduced from 5 to 2 using AT. A 3-session program of AT was carried out daily, and the pain was alleviated enough to carry out daily activities.

## 3. Discussion

Nerve damage due to implants can be transient or permanent; this finding will depend on the cause and extent of the injury. Nerve wounding may result in numbness, allodynia, paresthesia, or dysesthesia. Renton et al. reported that in a population with many trained operators, 8 of 1012 patients with an implant placement developed trigeminal neuropathy, with a prevalence of 0.8% [11]. To date, only two studies have investigated QST after an implant-induced nerve injury [12,13]. Hartmann et al. performed a QST for extra-oral and intra-oral use in patients with implant placements and in patients suffering from impaired neurological function due to implant placement and numbness. They found abnormal sensory responses to touch coexisting with temperature nociception. To our knowledge, this is the first report of a comprehensive somatosensory profile in a patient with PTTN. Our patient exhibited mechanical hyperalgesia (MDT = 0.65 mN) and heat hyposensitivity (WDT = 15.6 °C) in the mandibular region on the affected side. The activation of Aβ nerve fibers may lead to neuropathic pain when implant surgery causes a nerve injury. Therefore, non-painful mechanical stimulation, normally mediated by Aβ fibers, can lead to painful sensations. Under physiological conditions, mechanosensory sensations are transmitted by large, myelinated A fibers, and interoceptive (homeostatic) sensations (including pain, itch, temperature, chemical sensation, and pleasant touch) are transmitted by small-diameter afferent nerves. These two phylogenetically distinct systems have intermingled throughout evolution, communicating with each other, contributing to global body cognition [14], segregating pathways at the spinal/trigeminal level, and enhancing sensory specificity.

The successful management of persistent neuropathic pain in the orofacial region remains challenging. The response to pharmacological treatment is approximately 11%, and according to a long-term follow-up study, only one-third of patients experienced meaningful improvements [15]. AT is frequently used as a therapeutic approach for multimodal pain therapy. Relaxation is suggested to affect pain by reducing tissue oxygen requirement and degrading lactic acid, relieving skeletal muscle tension and anxiety, and releasing endorphins [6]. The efficacy of AT in individuals suffering from pain has been investigated in numerous randomized controlled trials. A meta-analysis showed that AT can be used as an effective relaxation technique for individuals suffering from chronic pain. Although AT was effective in this patient who could not use pharmacological therapy, it

may also be effective for chronic pain, such as chronic low back pain, arthritis pain, and chronic orofacial pain. However, the results of this case study suggest that a clinically significant treatment response is possible after AT.

Since implant treatment takes a long time and the cost is high, the failure of implant treatment leads to patient injustice, which may result in litigation. Injured patients with chronic pain not only perceive injustice but also catastrophize their experience of pain. Yamada et al. investigated the association between demographic variables (duration of pain, cause of injury, liability for injury, employment status, compensation, and dispute) regarding injury-related pain and IEQ-J scores to reinforce the evidence from previous studies [10]. Presumably, victims injured by another's error or negligence and the injured who are compensated are more likely to perceive injustice than those with a self-inflicted injury or without compensation. Sullivan et al. indicated that motor vehicle accidents are associated with higher IEQ scores [7]. Ferrari reported that whiplash victims, six months post-injury, had higher IEQ scores than those three months post-injury [16]. In patients with sub-acute or chronic whiplash-associated disorders, the IEQ scores are associated with pain severity, displays of pain behavior, depressive symptoms, work disability, and post-traumatic stress symptoms [17]. Hayashi et al. reported that the appropriate cutoff score for the IEQ would be a score of 21 or higher when related to the long-term disability of patients with a whiplash-associated disorder [17]. In this case, the IEQ score was relatively high (26). The reasons for this include the high cost of implant treatment, the treatment period of one year and three months, and the fact that the risks of implant treatment were not explained before surgery. In the future, it will be necessary to investigate the relationship between IEQ and factors such as pain intensity, QST parameters, the severity of neuropathy, and treatment costs in clinical studies.

## 4. Conclusions

Post-implant nerve injury may increase perceived patient injustice after surgery. Perceived injustice in elderly patients may prolong chronic pain; therefore, an adequate preoperative consultation is required.

**Author Contributions:** The contribution of the authors is as follows. N.N.: contributed to the concept, design, analysis, and interpretation, and drafted and critically revised the manuscript; K.T.: contributed to the investigation; K.O., J.N., and S.T.: contributed to the treatment of the patient; A.O.-o. and Y.K.: contributed to the concept, design, analysis, and interpretation, and drafted and critically revised the manuscript All the authors gave their final approval and agreed to be accountable for all aspects of the work. All authors have read and agreed to the published version of the manuscript.

**Funding:** This study was supported in part by research grants from the Dental Research Center of Nihon University School of Dentistry.

**Institutional Review Board Statement:** This study was approved by the Ethics Committee of the Nihon University School of Dentistry (EP16 D021) and was conducted in accordance with the principles of the Declaration of Helsinki. Written informed consent was obtained from all the participants.

**Informed Consent Statement:** Written informed consent was obtained from the patient to publish this paper.

**Data Availability Statement:** Not applicable.

**Acknowledgments:** We thank all our colleagues for their helpful discussions.

**Conflicts of Interest:** The authors declare no conflict of interest.

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
