# Peer review of "Post-Traumatic Trigeminal Neuropathic Pain after Dental Implant Surgery and the Injustice Experience Questionnaire"

_2035-8377, doi:10.3390/neurolint15010007_

Round 1

Reviewer 1 Report

Line 21 two dots

Line 34-35 there are more specific pharmacologic protocols for neuropathic pain

Did you use the Injustice Experience Questionnaire (IEQ) in the original language or translated? Is there a cross-cultural validated form in your language?

Line 65 hot soba noodles ... could you please describe the consistency?

Line 74-75 Simple neurosensory testing ... could you please explain better

Line 109-110  The patient’s total IEQ score 109 was 26/48. Could you please discuss this result?

Author Response

Reviewer 1

Comments and Suggestions for Authors

We corrected following suggetion.

Line 21 two dots

Line 34-35 there are more specific pharmacologic protocols for neuropathic pain

Did you use the Injustice Experience Questionnaire (IEQ) in the original language or translated? Is there a cross-cultural validated form in your language?

We added following statement.

The IEQ’s translation into Japanese was based on guidelines for cross-cultural adaptation of self-report measures. We employed the Japanese version of the IEQ.

Line 65 hot soba noodles ... could you please describe the consistency?

At first, patient felt only numbness, but gradually became aware of pain and discomfort from hot soba noodles, result in development of possible neuropathic condition.

Line 74-75 Simple neurosensory testing ... could you please explain better

We added semiquantitative chairside techniques.

Line 109-110  The patient’s total IEQ score 109 was 26/48. Could you please discuss this result?

We added following statement.

Hayashi et al. reported that the appropriate cutoff score for the IEQ would be a score of 21 or higher, when related to the long-term disability of patients with acute whiplash-associated disorder.

Reviewer 2 Report

The introduction and Discussion should be improved.

Minor adjustments.

This part should have modification:

The patient’s pain was evaluated using the IEQ, which was translated and back- 99 translated into English and Japanese [10]. IEQ is a self-report measure of injury-related 100 perceived injustice. The IEQ consists of 12 statements assessing the responder's appraisals 101 of their injury and its consequences in terms of unfairness (e.g., “It all seems so unfair”), 102 irreparability of loss (“My life will never be the same”), and attribution of blame (“I am 103 suffering because of someone else's negligence”). Responders rated the frequency with 104 which they had experienced each of the 12 pain-related perceptions. The IEQ uses a 5- 105 point scale from 0 to 4 as follows: never (0), rarely (1), sometimes (2), often (3), and all the 106 time (4). 

This part should have modification:

Reports of nerve damage due to implants revealed that the incidence rate is close to 119 40% immediately after surgery and that the residual disability rate is close to 7% after 120 more than a year.

How this number is estimated? Data about Europe, and locations of the World shoul be cited.

Author Response

Reviewer 2

Comments and Suggestions for Authors

The introduction and Discussion should be improved.

Minor adjustments.

This part should have modification:

The patient’s pain was evaluated using the IEQ, which was translated and back- 99 translated into English and Japanese [10]. IEQ is a self-report measure of injury-related 100 perceived injustice. The IEQ consists of 12 statements assessing the responder's appraisals 101 of their injury and its consequences in terms of unfairness (e.g., “It all seems so unfair”), 102 irreparability of loss (“My life will never be the same”), and attribution of blame (“I am 103 suffering because of someone else's negligence”). Responders rated the frequency with 104 which they had experienced each of the 12 pain-related perceptions. The IEQ uses a 5- 105 point scale from 0 to 4 as follows: never (0), rarely (1), sometimes (2), often (3), and all the 106 time (4).

We changed to following sentence.

The patient’s pain was evaluated using the IEQ, which was translated and back-translated into English and Japanese [10]. The IEQ’s translation into Japanese was based on guidelines for cross-cultural adaptation of self-report measures. We employed the Japanese version of the IEQ. IEQ is a self-report measure of injury-related perceived injustice. The IEQ consists of 12 statements assessing the responder's appraisals of their injury and its consequences in terms of unfairness (e.g., “It all seems so unfair”), irrepara-bility of loss (“My life will never be the same”), and attribution of blame (“I am suffering because of someone else's negligence”). The patient assessed the frequency with which they had experienced each of the 12 pain-related perceptions. The IEQ employs a 5-point scale from 0 to 4 as follows: never (0), rarely (1), sometimes (2), often (3), and all the time (4). The total score was calculated by adding all the items (theoretical range, 0-48). In addi-tion, the proportion of patient who endorsed each item was calculated. Endorsement was defined as a rating of ‘sometimes’ or greater (≥2). The patient’s total IEQ score was 26/48.

This part should have modification:

Reports of nerve damage due to implants revealed that the incidence rate is close to 119 40% immediately after surgery and that the residual disability rate is close to 7% after 120 more than a year.

How this number is estimated? Data about Europe, and locations of the World shoul be cited.

We changed to following sentence.

Nerve damage due to implants can be transient or permanent; this finding will depend on the cause and extent of the injury. Nerve wounding may result numbness, allodynia, paresthesia, or dysesthesia.
